# FEDERATED LEARNING FOR TIME-SERIES HEALTHCARE SENSING WITH INCOMPLETE MODALITIES

## ABSTRACT

Many healthcare sensing applications utilize multimodal time-series data from sensors embedded in mobile and wearable devices. Federated Learning (FL), with its privacy-preserving advantages, is particularly well-suited for health applications. However, most multimodal FL methods assume the availability of complete modality data for local training, which is often unrealistic. Moreover, recent approaches tackling incomplete modalities scale poorly and become inefficient as the number of modalities increases. To address these limitations, we propose `FLISM`, an efficient FL training algorithm with incomplete sensing modalities while maintaining high accuracy. `FLISM` employs three key techniques: (1) modality-invariant representation learning to extract effective features from clients with a diverse set of modalities, (2) modality quality-aware aggregation to prioritize contributions from clients with higher-quality modality data, and (3) global-aligned knowledge distillation to reduce local update shifts caused by modality differences. Extensive experiments on real-world datasets show that `FLISM` not only achieves high accuracy but is also faster and more efficient compared with state-of-the-art methods handling incomplete modality problems in FL.

## 1 INTRODUCTION

In healthcare sensing, many applications leverage multimodal time-series data from an array of sensors embedded in mobile and wearable devices (Ramachandram & Taylor, 2017). For example, mobile devices equipped with motion and physiological sensors capture multimodal data to detect eating episodes (Shin et al., 2022), monitor physical activities (Reiss & Stricker, 2012), track emotional states (Park et al., 2020), and assess stress levels (Schmidt et al., 2018). Thanks to its privacy-preserving characteristics, Federated Learning (FL) (McMahan et al., 2017) is particularly suited for these applications, supporting local model training without sharing raw data with a central server. Despite this benefit, FL encounters challenges involving multimodal health sensing data.

One major problem is incomplete modalities (Vaizman et al., 2018; Feng & Narayanan, 2019; Li et al., 2020a) where factors such as limited battery life, poor network connections, and sensor malfunctions prevent users from utilizing all modalities for local training, leading to variations in the multimodal data availability across FL clients. In centralized machine learning, this issue is often addressed using statistical techniques (Yu et al., 2020) or deep learning-based imputation methods (Zhao et al., 2022; Zhang et al., 2023). However, the privacy-preserving nature of FL limits the direct exchange of raw data between clients and the server, making it difficult to apply existing approaches in an FL setting.

A way to handle incomplete modalities in FL is to train separate encoders for each available modality during local client training and use the extracted features from these encoders to train a multimodal fusion model. This way of training, known as *intermediate fusion*, has been widely adopted in recent studies that address incomplete modalities in FL (Feng et al., 2023; Ouyang et al., 2023). Another approach uses *deep imputation* (Zheng et al., 2023), where cross-modality transfer models trained on complete data are used to impute missing modalities. Although these approaches offer flexibility in adapting to varying modalities, they suffer from high communication and computation costs, limiting their scalability as the number of modalities increases. As discussed in Appendix A.1.1, this lack of scalability is a significant challenge in multimodal healthcare sensing FL, where the variety of personal devices and sensors continues to grow. Experiment results in Appendices A.1.2

and A.1.3 confirm that current approaches, including intermediate fusion and deep imputation, scale poorly and remain resource-inefficient.

Unlike intermediate fusion, *early fusion* combines multimodal streams early at the input level. It is particularly well-suited for multimodal time-series healthcare sensing applications, as it captures intricate modality relationships (Pawłowski et al., 2023) and enhances efficiency by training a single model (Snoek et al., 2005). However, the standard method of imputing missing modalities in early fusion using raw statistics is not feasible in FL setting, as the server cannot access clients' raw data. This restriction leaves zero imputation as the only option. Nevertheless, without properly addressing incomplete modalities, zero imputation alone leads to distribution drifts and significant performance drops, as evidenced in Appendix A.2.

We present FLISM (Federated Learning with Incomplete Sensing Modalities), an efficient FL algorithm for multimodal time-series healthcare sensing tasks with incomplete modalities. Our goal is to leverage the efficiency of early fusion while maintaining high accuracy. Achieving this is challenging due to several factors: (1) early fusion with zero imputation alone can cause the model to learn biased relationships between modalities; (2) the quality of local modality data varies across clients, resulting in a suboptimal global model, and (3) clients, especially those with limited modalities, may deviate significantly after local updates. To overcome these challenges, we propose three key techniques: modality-invariant representation learning to extract robust features from diverse modalities in early fusion, tackling (1); modality quality-aware aggregation to prioritize updates from clients with higher-quality modality data, addressing (2); and global-aligned knowledge distillation to reduce the impact of drifted local updates, solving (3).

We conducted extensive experiments and evaluated the performance of FLISM against six baselines using four real-world multimodal time-series healthcare sensing datasets. The results demonstrate the effectiveness of our method in both accuracy and efficiency. FLISM improves accuracy across all early fusion baselines, with F1 score gains from .043 to .143, and outperforms intermediate fusion methods with F1 score improvements from .037 to .055, while being $3.11\times$ faster in communication and $2.14\times$ more efficient in computation. FLISM also surpasses deep imputation methods, achieving a .073 F1 score increase and reducing communication and computational costs by $77.50\times$ and $35.30\times$, respectively.

## 2 RELATED WORK

**Multimodal Healthcare Sensing.** Multimodal learning is becoming more prevalent in healthcare sensing, with applications ranging from physical activity tracking (Reiss & Stricker, 2012) and eating episode detection (Shin et al., 2022) to emotion and stress assessment (Park et al., 2020; Yu & Sano, 2023). To effectively utilize multimodal data, recent studies have proposed advanced training methods, including enhancing data representations by optimizing cross-correlation (Deldari et al., 2022) and incorporating self-supervised learning to build foundational model for healthcare sensing tasks (Abbaspourazad et al., 2024). However, transmitting raw health data to a centralized server raises significant privacy concerns.

**Federated Learning.** Federated Learning (FL) (McMahan et al., 2017) offers a promising solution for healthcare applications, as it enables local training on client devices without the need to send sensitive raw data to a central server. However, most existing approaches (Xiong et al., 2022; Le et al., 2023) assume complete modality data for local training, which is often unrealistic due to factors such as battery constraints, network issues, and sensor malfunctions. In contrast, FLISM works flexibly with incomplete modalities across various multimodal healthcare sensing tasks.

**FL with Incomplete Modalities.** Research in multimodal FL with incomplete modalities has recently gained attention. For instance, Ouyang et al. (2023) uses a two-stage training framework, while Zheng et al. (2023) introduces an autoencoder-based method to impute missing modalities. Feng et al. (2023) uses attention-based fusion to integrate outputs from separately trained uni-modal models. Although these methods demonstrate effectiveness across various tasks, they often encounter challenges with efficiency and scalability as the number of modalities grows. In contrast, FLISM utilizes early fusion to enhance efficiency while incorporating techniques to maintain high accuracy.

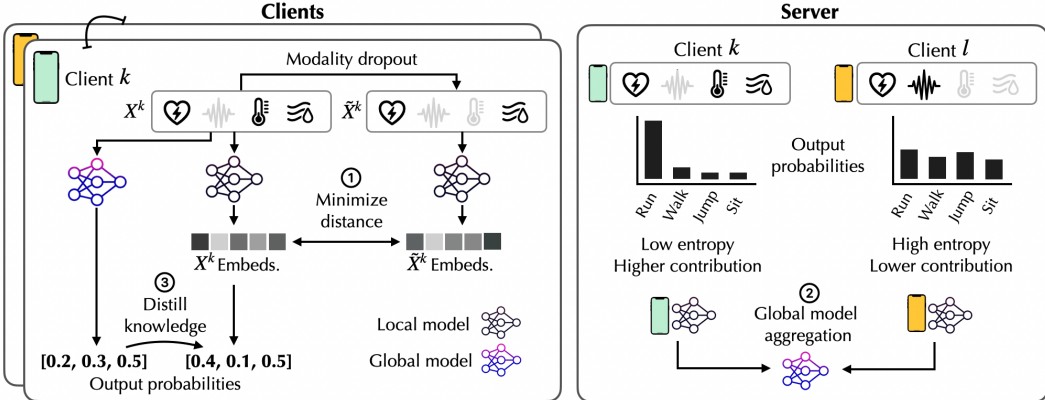

Figure 1: Overview of FLISM, consisting of three key components: ①: Modality-Invariant Representation Learning (MIRL) learns to extract effective features, ②: Modality Quality-Aware Aggregation (MQAA) priorities clients with higher-quality modality data, and ③: Global-Aligned Knowledge Distillation (GAKD) reduces deviations in client updates by aligning the predictions of the local model with that of the global.

## 3 METHOD

### 3.1 PROBLEM SETTING

Consider a multimodal time-series healthcare sensing task in FL involving $M \geq 2$ modalities with $K$ participating clients. Each client $k$ has a local training dataset $D_k = \{(x_{i,1}^k, \ldots, x_{i,m_k}^k, y_i^k)\}_{i=1}^{n_k}$ of size $n_k = |D_k|$ with $m_k$ modalities, where $x_{i,j}^k$ is the data from the $j$-th modality for the $i$-th sample, and $y_i^k$ is the corresponding label. The global objective (McMahan et al., 2017) is to minimize the local objectives from clients:

$$\min_w F(w) \coloneqq \sum_{k=1}^{K} \frac{n_k}{n} F_k(w), \tag{1}$$

where $n = \sum_{k=1}^{K} n_k$, and the local objective for a client $k$ is defined as:

$$F_k(w) = \frac{1}{n_k} \sum_{i=1}^{n_k} f\left(w; x_{i,1}^k, \ldots, x_{i,m_k}^k, y_i^k\right). \tag{2}$$

Here, $f(w; x_{i,1}^k, \ldots, x_{i,m_k}^k, y_i^k)$ is the loss for a single data point, with model $f$ parameterized by $w$.

### 3.2 THE FLISM APPROACH

We introduce FLISM, an efficient FL algorithm for multimodal time-series healthcare sensing with incomplete modalities (overview shown in Figure 1). FLISM combines the resource efficiency of early fusion with high accuracy by addressing three key challenges. First, early fusion with zero imputation can lead to biased modality relationships. We solve this problem with Modality-Invariant Representation Learning (MIRL, §3.2.1), which extracts robust features from incomplete modality data. Second, the quality of modality data differs among clients, and simply aggregating their updates can result in a suboptimal global model. To counter this, we propose Modality Quality-Aware Aggregation (MQAA, §3.2.2), which prioritizes contributions from clients with higher-quality modalities. Finally, clients with limited or lower-quality data may experience significant deviations during local updates. We address this issue with Global-Aligned Knowledge Distillation (GAKD, §3.2.3), which aligns local model predictions with that of the global model to minimize update drift. Complete pseudocode of FLISM is given in Algorithm 1.

### 3.2.1 Modality-Invariant Representation Learning

Early fusion with zero imputation alone can learn biased modality relationships, leading to performance degradation (supporting experimental results are given in Appendix A.2). To address this problem, we propose Modality-Invariant Representation Learning (MIRL, Figure 1–①), a technique that extracts effective features regardless of available modalities. Formally, let us consider two samples $\boldsymbol{x}_i$ and $\boldsymbol{x}_j$, each with $1 \leq m_i, m_j \leq M$ modalities:

$$\boldsymbol{x}_i = (x_{i,1} \ldots x_{i,m_i}), \quad \boldsymbol{x}_j = (x_{j,1} \ldots x_{j,m_j}).$$

Our goal is to learn a function $f : X \rightarrow H$ that maps samples with varying modality combinations into an embedding space $H$. To achieve this goal, we employ supervised contrastive learning (SupCon) (Khosla et al., 2020). SupCon leverages label information to cluster embeddings of samples sharing the same label and separate those with different labels. We employ SupCon to position samples sharing the same label $y$ close together in $H$, regardless of their available modalities:

$$d\big(f(\boldsymbol{x}_i), f(\boldsymbol{x}_j)\big)\big|_{y_i \neq y_j} > d\big(f(\boldsymbol{x}_i), f(\boldsymbol{x}_j)\big)\big|_{y_i = y_j} \tag{3}$$

where $d$ is a distance metric, $d : H \times H \rightarrow \mathbb{R}^+$. Unlike conventional SupCon that uses image-based augmentations such as cropping and flipping, we adapt it for multimodal time-series sensing data by generating augmented samples through random modality dropout and noise addition. This adaptation enables the model to learn modality-invariant representations, ensuring that embeddings remain consistent for samples with the same label despite varying input modalities. Specifically, consider a client $k$ with $m_k$ modalities available for local training, where $2 \leq m_k \leq M$.[1] For each sample $\boldsymbol{x}_i^k$ in client's dataset, we generate an augmented sample $\tilde{\boldsymbol{x}}_i^k$ by randomly dropping up to $m_k - 1$ modalities and perturbing the data by adding noise:

$$\tilde{\boldsymbol{x}}_i^k = \texttt{ModalityDropout}(\boldsymbol{x}_i^k, \mathcal{M}^k) + \epsilon_i, \tag{4}$$

where $\texttt{ModalityDropout}$ randomly selects a subset $\mathcal{M}^k \subseteq \{1, \ldots, m_k\}$ of modalities to retain, with $1 \leq |\mathcal{M}^k| \leq m_k - 1$, and sets modalities not in $\mathcal{M}^k$ to zero. The noise term $\epsilon_i$ is sampled from a Gaussian distribution $\epsilon_i \sim \mathcal{N}(\mu, \sigma^2)$.

Within a training batch $B$ containing $|B|$ samples, we combine the original samples $\boldsymbol{x}_i^k$ and their augmented counterparts $\tilde{\boldsymbol{x}}_i^k$ to form an expanded batch $\{\boldsymbol{x}_j^k, y_j^k\}$ for $j \in J = 1, \ldots, 2|B|$. Here, each $\boldsymbol{x}_j^k$ is either an original sample $\boldsymbol{x}_i^k$ or an augmented sample $\tilde{\boldsymbol{x}}_i^k$.

We define an encoder $h : X \rightarrow \mathbb{R}^{d_{enc}}$ and a projection head $g : \mathbb{R}^{d_{enc}} \rightarrow H$, where $H \subseteq \mathbb{R}^{d_{proj}}$ is the feature embedding space. The overall mapping function $f : X \rightarrow H$ is defined as $f(x) = g(h(x)) = z$, where $z \in H$. The supervised contrastive loss (Khosla et al., 2020) is then defined as:

$$L_{SC} = \sum_{j \in J} \frac{-1}{|P(j)|} \sum_{p \in P(j)} \log\Big(\frac{exp(z_j \cdot z_p / \tau)}{\sum_{q \in Q(j)} exp(z_j \cdot z_q / \tau)}\Big). \tag{5}$$

In this context, $z_j = f(x_j) = g(h(x_j))$ is the embedding of sample $x_j$ and $\tau$ is a temperature parameter. $Q(j) \equiv J \setminus \{j\}$, $P(j) \equiv \{p \in Q(j) : y_p = y_j\}$ includes the indices of all positive pairs in the batch, distinct from $j$.

In summary, MIRL reduces modality bias by learning effective embeddings invariant to the available input modalities, as demonstrated by results in Appendix B.

### 3.2.2 Modality Quality-Aware Aggregation

In multimodal time-series health sensing, certain modalities influence performance more than others. For example, electrodermal activity and skin temperature may provide more informative data than accelerometer readings for stress detection, whereas accelerometer data is more crucial than other sensors for human activity recognition. Similarly, clients have varying sets of available modalities to perform FL; some possess higher-quality or more informative modalities, while others have limited or lower-quality modalities. Consequently, it is essential to prioritize updates from clients with more

---

[1]To employ modality dropout, a client must have at least two modalities available.

informative modalities to enhance the global model. To achieve this, we introduce Modality Quality-Aware Aggregation (MQAA, Figure 1–②), an aggregation technique that gives greater weight to updates from clients with higher-quality modality data.

We hypothesize that if a client has more complete and higher modality data, the client model would produce more reliable and confident predictions. To quantify this prediction certainty, we employ the entropy metric $H(X)$, which measures the uncertainty of a random variable $X$ (Shannon, 1948). Specifically, we found that lower entropy values, indicating higher confidence predictions, reflect the quality and completeness of available modality data (Appendix C). Building on this finding, we propose leveraging entropy to evaluate client updates, allowing clients with more informative modalities to exert a greater influence on the global model.

Formally, let $S_t$ denote a subset of clients selected in a training round $t \in [T]$. Each client $k \in S_t$ performs a local update and calculates the entropy of the updated model predictions on its local private training data $X^k$, which contains $n_k$ samples in a task with $|C|$ classes. Recognizing that lower entropy corresponds to higher-quality client updates, we define weight $r_t^k$ assigned to each client $k$ in the global update as the inverse of its average entropy $(H(X^k))^{-1}$:

$$r_t^k = (-\frac{1}{n_k} \sum_{i=1}^{n_k} \sum_{j=1}^{|C|} p_{i,j} \log p_{i,j})^{-1}. \qquad (6)$$

The global model is then updated as follows:

$$W_t \leftarrow \sum_{k \in S_t} \frac{r_t^k}{r_t} w_t^k, \qquad (7)$$

where $p_{i,j}$ is the prediction probability of a model $w_t^k$ for sample $i$ and class $j$, and $r_t = \sum_{k \in S_t} r_t^k$.

In essence, MQAA strategically amplifies the impact of high-quality client updates, resulting in a more accurate and reliable global model. In Appendix F, we discuss future work, including possible extensions to MQAA for challenging scenarios with limited high quality client updates.

### 3.2.3 GLOBAL-ALIGNED KNOWLEDGE DISTILLATION

When clients have less informative or limited modality sets, their local model updates can become significantly biased (Karimireddy et al., 2020). This bias can degrade the global model's performance and its ability to generalize across different sets of modalities, especially when training rounds involve clients with lower-quality local modality data. To mitigate the impact of these biased updates, we employ Global-Aligned Knowledge Distillation (GAKD, Figure 1–③). GAKD leverages knowledge distillation (KD) (Hinton et al., 2015) to reduce the influence of less informative modality data on the global model. Specifically, during local training we distill knowledge from the global model to ensure that local model predictions align closely with those of the global model. This is because the global model contains a more comprehensive and generalized knowledge as it aggregates diverse data from all clients across various modalities.

The goal is to minimize the difference between predictions of the local model $F_k$ parameterized by $w_t^k$, and those of the global model, parameterized by $W_{t-1}$:

$$\min_{w_t^k} \mathbb{E}_{\boldsymbol{x}^k \sim D_k}[F_k(W_{t-1}; \boldsymbol{x}^k) || F_k(w_t^k; \boldsymbol{x}^k)]. \qquad (8)$$

The distillation loss $L_{KD}$ can then be defined using the Kullback–Leibler Divergence (KLD) (Kullback & Leibler, 1951) between softened probability distributions of the global and local models:

$$L_{KD} = \frac{\tau^2}{n_k} \sum_{i=1}^{n_k} \text{KLD} \left[ \sigma \left( F_k(W_{t-1}; \boldsymbol{x}_i^k) \right) / \tau, \sigma \left( F_k(w_t^k; \boldsymbol{x}_i^k) \right) / \tau \right], \qquad (9)$$

where $\sigma$ is the softmax function applied on model logits, $\tau$ is the temperature parameter that smooths the probability distribution, and $n_k$ is the number of samples in the local dataset $D_k$ of a client $k$.

Ultimately, GAKD effectively reduces modality-induced client biases, enhancing the global model's performance and its ability to generalize across diverse modality data.

---

**Algorithm 1** `FLISM` Algorithm

---

**Input:** Number of clients $K$ indexed by $k$, number of training rounds $T$, number of local training epochs $E$, fraction of clients $p$ selected to perform training. Each client has local training dataset $D_k$ with $m_k$ modalities. $\gamma$ is the hyperparameter in knowledge distillation.

1: **Server Executes:**
2:     initialize the global model $W_0$
3:     **for** global round $t = 1, \ldots, T$ **do**
4:         $S_t \leftarrow$ select random set of $\max(p \cdot K, 1)$ clients
5:         **for** each client $k \in S_t$ in parallel **do**
6:             $w_t^k, r_t^k \leftarrow \text{ClientUpdate}(k, W_{t-1})$
7:         **end for**
8:         $r_t \leftarrow \sum_{k \in S_t} r_t^k$
9:         $W_t \leftarrow \sum_{k \in S_t} \frac{r_t^k}{r_t} \cdot w_t^k$                      ▷ Modality quality-aware aggregation
10:     **end for**
11: **ClientUpdate**$(k, W_{t-1})$:
12:     $w_t^k \leftarrow W_{t-1}$
13:     $B \leftarrow$ split $D_k$ into batches of size $|B|$
14:     $\mathcal{M}_t^k \subseteq \{1, \ldots, m_k\} \leftarrow$ set of modalities to retain, used in Equation 4
15:     **for** each local epoch $e = 1, \ldots, E$ **do**
16:         **for** each batch $b \in B$ **do**
17:             $\tilde{b} \leftarrow \text{ModalityDropout}(b, \mathcal{M}_t^k) + \mathcal{N}(\mu, \sigma^2)$              ▷ Equation 4
18:             $L_{SC} \leftarrow \text{MIRL}(b, \tilde{b}, w_t^k)$          ▷ Modality-invariant representation learning
19:             $L_{KD} \leftarrow \text{GAKD}(b, W_{t-1}, w_t^k)$         ▷ Global-aligned knowledge distillation
20:             $L_{CE} \leftarrow \text{CrossEntropy}(b, w_t^k)$            ▷ Classification loss
21:             $L_{Total} \leftarrow L_{SC} + \gamma L_{KD} + L_{CE}$
22:             $w_t^k \leftarrow \text{SGD}(w_t^k, L_{Total})$
23:         **end for**
24:     **end for**
25:     $r_t^k \leftarrow \text{Entropy}(D_k, w_t^k)^{-1}$                              ▷ Equation 6
26:     **return** $w_t^k, r_t^k$

---

## 4 RESULTS

### 4.1 EXPERIMENTS

**Datasets.** We use four publicly available multimodal time-series healthcare sensing datasets in our experiments: PAMAP2 (Reiss & Stricker, 2012), WESAD (Schmidt et al., 2018), RealWorld HAR (Sztyler & Stuckenschmidt, 2016) (abbreviated as RealWorld), and Sleep-EDF (Goldberger et al., 2000; Kemp et al., 2000).

**Baselines.** We compare `FLISM` with the following baselines, including three *early fusion* baselines: 1) FedAvg (McMahan et al., 2017), 2) FedProx (Li et al., 2020b), 3) MOON (Li et al., 2021); two *intermediate fusion* methods designed to address incomplete modality problem in FL: 4) FedMulti-Modal (Feng et al., 2023) (abbreviated as FedMM), 5) Harmony (Ouyang et al., 2023); and a *deep imputation* approach: 6) AutoFed (Zheng et al., 2023).

Datasets, baseline descriptions, and implementation details are provided in Appendix D.

### 4.2 ACCURACY ANALYSIS

We conducted experiments under various incomplete modality scenarios to evaluate `FLISM`'s accuracy compared with the baselines. The results are shown in Table 1. Additional analysis with complete modalities is provided in Appendix E. The final test accuracy is measured using the macro F1 score ($F1$), recommended for imbalanced data (Plötz, 2021). `FLISM`'s improvements are denoted as $\Delta F1$. A higher value of $p$ signifies an increased incidence of clients with incomplete modalities.

Table 1: Accuracy improvement of `FLISM` over baselines with various incomplete modality ratios. **EF** denotes Early Fusion, whereas **IF** represents Intermediate Fusion.

| Method | FedAvg (EF) | | FedProx (EF) | | MOON (EF) | | FedMM (IF) | | Harmony (IF) | |
|---|---|---|---|---|---|---|---|---|---|---|
| **Accuracy** | F1 | Δ F1 | F1 | Δ F1 | F1 | Δ F1 | F1 | Δ F1 | F1 | Δ F1 |
| | | | | | *p =40%* | | | | | |
| PAMAP2 | .730 | .076 ↑ | .732 | .074 ↑ | .732 | .074 ↑ | .758 | .048 ↑ | .744 | .062 ↑ |
| WESAD | .672 | .020 ↑ | .672 | .020 ↑ | .438 | .254 ↑ | .556 | .136 ↑ | .656 | .036 ↑ |
| RealWorld | .796 | .014 ↑ | .792 | .018 ↑ | .660 | .150 ↑ | .804 | .006 ↑ | .782 | .028 ↑ |
| Sleep-EDF | .706 | .008 ↑ | .704 | .010 ↑ | .680 | .034 ↑ | .686 | .028 ↑ | .554 | .160 ↑ |
| | | | | | *p =60%* | | | | | |
| PAMAP2 | .740 | .050 ↑ | .740 | .050 ↑ | .738 | .052 ↑ | .750 | .040 ↑ | .710 | .080 ↑ |
| WESAD | .610 | .056 ↑ | .608 | .058 ↑ | .380 | .286 ↑ | .516 | .150 ↑ | .648 | .018 ↑ |
| RealWorld | .728 | .036 ↑ | .724 | .040 ↑ | .588 | .176 ↑ | .796 | .032 ↓ | .768 | .004 ↓ |
| Sleep-EDF | .698 | .020 ↑ | .698 | .020 ↑ | .574 | .144 ↑ | .680 | .038 ↑ | .538 | .180 ↑ |
| | | | | | *p =80%* | | | | | |
| PAMAP2 | .678 | .062 ↑ | .680 | .060 ↑ | .696 | .044 ↑ | .740 | .000 - | .704 | .036 ↑ |
| WESAD | .504 | .056 ↑ | .506 | .054 ↑ | .372 | .188 ↑ | .508 | .052 ↑ | .630 | .070 ↓ |
| RealWorld | .626 | .098 ↑ | .628 | .096 ↑ | .584 | .140 ↑ | .770 | .046 ↓ | .782 | .058 ↓ |
| Sleep-EDF | .680 | .020 ↑ | .680 | .020 ↑ | .522 | .178 ↑ | .676 | .024 ↑ | .512 | .188 ↑ |
| | | | | | *Averaged* | | | | | |
| PAMAP2 | .716 | .063 ↑ | .717 | .061 ↑ | .722 | .057 ↑ | .749 | .029 ↑ | .719 | .059 ↑ |
| WESAD | .595 | .044 ↑ | .595 | .044 ↑ | .397 | .243 ↑ | .527 | .113 ↑ | .645 | .005 ↓ |
| RealWorld | .717 | .049 ↑ | .715 | .051 ↑ | .611 | .155 ↑ | .790 | .024 ↓ | .777 | .011 ↓ |
| Sleep-EDF | .695 | .016 ↑ | .694 | .017 ↑ | .592 | .119 ↑ | .681 | .030 ↑ | .535 | .176 ↑ |

`FLISM` achieves noticeable performance improvement over all baselines. It consistently outperforms all early fusion methods, achieving average F1 score improvements of .043, .043, and .143 over FedAvg, FedProx, and MOON, respectively. FedAvg employs zero-imputation and conducts standard FL training. Although simple and efficient, zero-imputation distorts the data distribution, causing the model to learn incorrect relationships between modalities. Both FedProx and MOON incorporate techniques to handle clients with highly heterogeneous data, primarily addressing the standard non-IID (label skew) problem in FL. However, these methods perform similarly or even worse than the FedAvg when faced with incomplete modalities. This indicates that the data heterogeneity targeted by existing approaches differs from the challenges posed by incomplete modalities, rendering these techniques ineffective in such cases. In contrast, `FLISM` learns effective features with incomplete modalities and builds a more robust global model by prioritizing the clients with highly informative modality data, resulting in enhanced accuracy.

Compared with the intermediate fusion algorithms, FedMM and Harmony, `FLISM` shows average F1 score improvements of .037 and .055, respectively. Note that FedMM and Harmony perform similarly or slightly better than `FLISM` on datasets with complementary modalities (e.g., RealWorld contains ten modalities collected from two unique sensors at five body locations). This is because intermediate fusion can leverage complementary embeddings during fusion, such as an accelerometer from the waist compensating for one from the chest. Nevertheless, intermediate fusion struggles with datasets that have more diverse and non-complementary modalities, such as Sleep-EDF and WESAD. In contrast, `FLISM` employs early fusion to capture relationships between modalities at an early stage and incorporates components specifically designed to handle incomplete modalities. Importantly, as an early fusion approach, `FLISM` is significantly more efficient in communication and computation than both Harmony and FedMM while maintaining similar or improved accuracy, as detailed next.

## 4.3 SYSTEM EFFICIENCY

We compare the communication and computation costs of `FLISM` with FedMM and Harmony, the state-of-the-art methods for handling incomplete modalities in FL. Communication cost is measured by the total time in exchanging model updates between the server and clients during FL training. Client upload and download speeds are sampled from FLASH (Yang et al., 2021), a simulation framework that contains hardware (communication and computation) capacities of 136K devices.

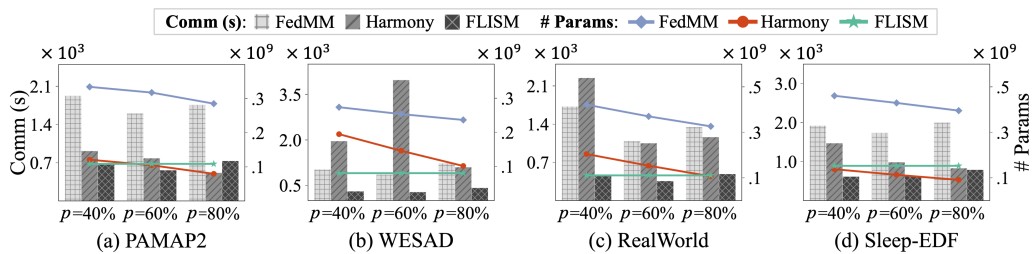

Figure 2: Comparison of `FLISM` with other baselines on communication and computation cost.

Computation cost is measured by the total number of model parameters trained by clients throughout the FL training process.

Figure 2 presents the results, where the x-axis denotes the incomplete modality ratio ($p$), the left y-axis indicates communication cost (in seconds), and the right y-axis represents the number of trained model parameters. Compared with `FLISM`, FedMM incurs $2.70\times\sim3.16\times$ higher communication overhead and is $2.81\times\sim3.36\times$ less computationally efficient. This is because each client must train and communicate a separate encoder for every modality in addition to the intermediate-fused classifier. Consequently, as the number of modalities increases, both the number of encoders and the associated overhead rise proportionally.

`FLISM` also surpasses Harmony by communicating model updates $1.13\times\sim7.01\times$ faster. Harmony's initial phase requires training multiple unimodal models, which significantly increases communication overhead as the number of modalities grows. This inefficiency also affects computational performance, making Harmony $1.40\times$ and $1.83\times$ less efficient on ten-modality RealWorld and WESAD datasets, respectively. For datasets with fewer modalities, such as PAMAP2 (six) and Sleep-EDF (five), Harmony's computational cost is comparable to or even better than `FLISM`. However, Harmony's F1 score declines on these datasets, particularly in Sleep-EDF where it records the lowest F1 score among all methods. In contrast, `FLISM` enhances resource efficiency by utilizing early fusion, eliminating the need to train separate unimodal models for each modality.

### 4.4 EVALUATION AGAINST A DEEP IMPUTATION APPROACH

Deep imputation methods, including AutoFed (Zheng et al., 2023), rely on a held-out complete-modality dataset to pretrain their imputation models, an unrealistic assumption in the FL setting. In contrast, our primary evaluations (§4.2∼§4.3) address more practical scenarios without requiring complete multimodal data. Therefore, we conducted separate experiments to fairly compare the performance of AutoFed with `FLISM`. AutoFed is designed for multimodal FL with only two modalities, requiring modifications for datasets with more than two modalities. Thus, we implemented AutoFed+, a variant capable of handling more than two modalities. Implementation details of these modifications are in Appendix D.4. For our comparative evaluation, we used the PAMAP2 (six modalities) and Sleep-EDF (five modalities) datasets. AutoFed requires training cross-modality imputation models for each unique pair of modalities, resulting in $M(M-1)$ models for a dataset with $M$ modalities. Using datasets with more modalities, such as those with ten, would necessitate an impractically large number of training models.

Table 2 presents the average F1 score, and communication and computation costs of `FLISM` and AutoFed+. `FLISM` outperforms AutoFed+ with average F1 score improvements of .078 and .067 for PAMAP2 and WESAD datasets, respectively.

Table 2: Comparison between AutoFed+ and `FLISM`.

| Method | AutoFed+ | | | FLISM | | |
|---|---|---|---|---|---|---|
| Dataset | F1 | Comm | Comp | F1 | Comm | Comp |
| PAMAP2 | 0.708 | 51.76K | 3.19B | **0.786** | **0.65K** | **0.11B** |
| Sleep-EDF | 0.659 | 51.42K | 6.31B | **0.726** | **0.68K** | **0.15B** |

The communication and computation costs for AutoFed+ stem primarily from the first phase, which involves pre-training generative imputation models for all $M(M-1)$ unique modality combinations. `FLISM` is $75.85\times\sim79.15\times$ more communication-efficient and incurs $29.31\times\sim41.28\times$ less computation overhead than AutoFed+, while consistently achieving higher F1 scores.

## 4.5 SCALABILITY ANALYSIS

As sensors become more integrated into healthcare devices, the demand for scalable multimodal FL systems increases (Appendix A.1.1). While our experimental datasets included up to ten sensing modalities, real-world applications may involve many more (Schmidt et al., 2018; Orzikulova et al., 2024). To better reflect these scenarios, we conducted scalability experiments comparing FLISM with the SOTA intermediate fusion methods, simulating healthcare sensing tasks with 5 to 30 modalities. We set the total number of clients to 100, with 10% participating in each FL training round. The incomplete modality ratio $p$, was set to 40%. The FL training lasted 20 rounds, with each client training their model for one local epoch per round.

The results, comparing FLISM with intermediate fusion baselines, are shown in Figure 3. The x-axis denotes the number of modalities and the y-axis indicates the communication and computation costs. The results show that for tasks involving five to thirty modalities, FedMM's communication cost increased by 2,743 seconds, while Harmony's rose by 25,847 seconds. Their computation costs also escalate, requiring 0.582B and 7.002B model parameters, respectively.

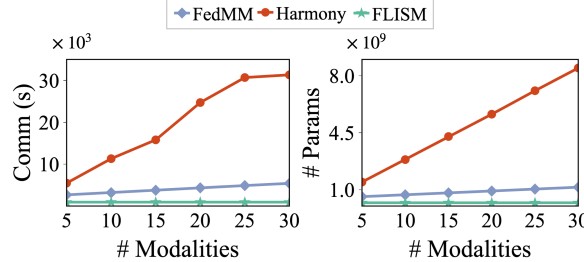

Figure 3: Scalability analysis of FLISM: Comparing communication (left) and computation (right) costs.

In contrast, FLISM maintains negligible system costs, and scales efficiently with more number of modalities. FLISM outperforms FedMM and Harmony with communication improvements between 2.89×∼5.83× and 5.91×∼33.74×, and in computation improvements between 2.86×∼5.74× and 7.33×∼42.07×, respectively.

## 4.6 ABLATION ANALYSIS

We conducted ablation analysis to assess the effectiveness of each component of FLISM. Table 3 shows the average F1 score across diverse incomplete modality ratios, maintaining consistency with the main experiments.

Table 3: Component-wise analysis of FLISM.

| Description | PAMAP2 | WESAD | RealWorld | Sleep-EDF |
|---|---|---|---|---|
| w/o MIRL, MQAA, GAKD | .716 | .595 | .717 | .695 |
| w/o MQAA, GAKD | .742 | .608 | .751 | .707 |
| w/o GAKD | .743 | .635 | .755 | .710 |
| **FLISM** | **.779** | **.639** | **.766** | **.711** |

The most basic version of FLISM, without any of three key components, is equivalent to FedAvg. Introducing modality-invariant representation learning (MIRL, §3.2.1) alone increases the average F1 by .021, indicating that learning to extract modality-invariant features enables the model to develop robust representations with incomplete modalities. Incorporating modality quality-aware adaptive aggregation (MQAA, §3.2.2) further boosts performance, adding .009 F1 improvement on top of the previous version. This improvement confirms that clients with different training modalities should contribute proportionally based on the quality of their modalities. Finally, integrating all three components, including global-aligned knowledge distillation (GAKD, §3.2.3), results in the complete version of FLISM. FLISM achieves the highest F1 score, showcasing the effectiveness of stabilizing drifted updates by aligning local model predictions with the global model. These results highlight each component's unique and significant contribution to the overall performance of FLISM.

## 5 CONCLUSION

We propose FLISM, an efficient FL algorithm for multimodal time-series healthcare sensing with incomplete modalities. FLISM extracts effective features through modality-invariant representation learning, adjusts the contribution of local updates by prioritizing clients with higher-quality modalities, and mitigates local update shifts caused by modality differences.

ETHICS STATEMENT

We have used publicly available multimodal time-series health sensing datasets in our experiments. There are no ethical issues with this paper.

REPRODUCIBILITY STATEMENT

We have provided the complete pseudocode of FLISM in Algorithm 1. Experimental and implementation details are included in Appendix D. Furthermore, anonymized source code is available at: https://anonymous.4open.science/r/IMFL-E113.

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

# A    MOTIVATIONAL EXPERIMENTS

We present the results of motivational experiments that illustrate the impact of incomplete modalities in healthcare sensing applications in FL settings. Our analysis shows that current approaches become increasingly ineffective, inefficient, and struggle to scale as the number of modalities grows.

## A.1    SCALABILITY AND EFFICIENCY CHALLENGES

### A.1.1    DEMAND FOR SCALABILITY IN MULTIMODAL HEALTHCARE SENSING FL

In healthcare sensing applications, leveraging various input modalities, from physiological sensors for emotional assessment (Kreibig, 2010), sleep tracking (Goldberger et al., 2000), to wearable devices for dietary monitoring (Merck et al., 2016; Bahador et al., 2021), enhances app performance (Ramachandram & Taylor, 2017). Additionally, there has been a recent expansion in both the variety of personal devices (such as smartwatches, bands, glasses, and rings) (Oura Ring, 2015; Apple Vision Pro, 2024) individuals own and in the spectrum of sensors (encompassing motion sensors and physiological sensors such as photoplethysmography (PPG) and electrodermal activity (EDA) sensors) (Schmidt et al., 2018) that are incorporated into these devices. This highlights the need for a scalable FL system to support multimodal healthcare sensing applications with tens of modalities. Yet, existing systems struggle to scale efficiently with an increase in the number of modality sources.

Below, we demonstrate that existing methods addressing missing modalities in FL such as *intermediate fusion* (Salehi et al., 2022; Xiong et al., 2022; Ouyang et al., 2023) and *deep imputation* (Zhao et al., 2022; Zheng et al., 2023), show limited scalability and inefficiency in resources and computation.

### A.1.2    RESOURCE COSTS OF INTERMEDIATE FUSION

Most existing works (Xiong et al., 2022; Salehi et al., 2022; Ouyang et al., 2023) tackle the missing modality problem in FL using *intermediate fusion*. In contrast to *early fusion*, where all sensing modalities are combined at the input stage to train a unified feature extractor and classifier, *intermediate fusion* trains individual unimodal feature extractors for each modality. This approach leads to significant scalability and efficiency challenges as the number of modalities in a multimodal task increases. To better assess the differences in efficiency and scalability, we compared the MACs (Multiply-Accumulate operations) and the number of trainable parameters between early and intermediate fusion mechanisms. We specifically focused on MACs and the number of model parameters because these metrics are key indicators of a model's resource consumption, including CPU, GPU, and memory usage, and therefore directly influencing model's practical deployment.

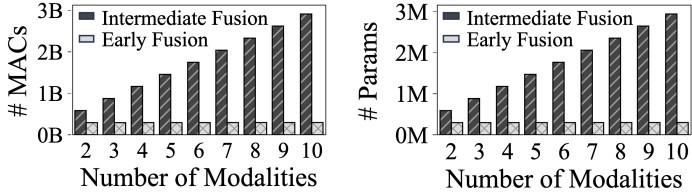

Figure 4: Comparison of resource usage between intermediate and early fusion based on the number of MACs (left) and model parameters (right).

Figure 4 shows the results. We used CNN+RNN-based model, a widely adopted architecture in multimodal sensing applications (Li et al., 2016; Ordóñez & Roggen, 2016; El-Sappagh et al., 2020).[2] We computed the MACs and trainable parameters for a two-second window with a sampling rate of 500Hz, utilizing the THOP toolkit (Zhu, 2021) compatible with PyTorch (Paszke et al., 2019). This analysis demonstrates that early fusion methods exhibit only a marginal increase in MACs and the number of trainable model parameters, which is nearly imperceptible compared to the linear esca-

---

[2]We also conducted experiments with CNN-based (LeCun et al., 1995; Zeng et al., 2014; Yang et al., 2015) models and observed similar trend.

lation seen in intermediate fusion approaches. This trend suggests that intermediate fusion is less scalable, particularly for multimodal sensing tasks that involve numerous modality sources.

### A.1.3 RESOURCE COSTS OF DEEP IMPUTATION

Recent multimodal FL studies explored deep imputation models such as autoencoders (Baldi, 2012) to address missing modalities (Zhao et al., 2022; Zheng et al., 2023). These works often assume that models capable of cross-modality transfer have been pre-trained with complete modality data before the primary task training. However, this assumption is often unrealistic in FL settings as the central server cannot access clients' raw complete modality data. Consequently, these cross-modality imputation models must be trained locally on each client. This requirement introduces additional training burdens, particularly for clients possessing more modalities.

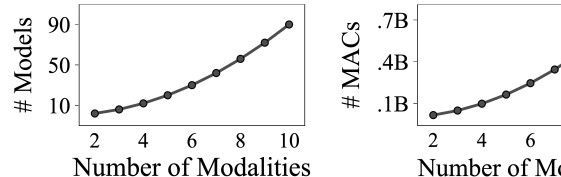

Figure 5: Deep imputation's extra training burden represented by the number of cross-modality transfer models (left) and the corresponding MACs (right).

Figure 5 shows the deep imputation's additional training cost represented by the number of cross-modality transfer models (left) and associated MACs (right). For instance, with two modalities A and B, a client must train two cross-modality transfer models: A to B and B to A. With three modalities, six combinations emerge: AB, AC, BA, BC, CA, CB. Since the permutations for $M$ modalities is $M \cdot (M-1)$, the complexity of the additional imputation model training increases quadratically. Considering the prevalent use of CNN-based blocks followed by transpose-CNN blocks (Zheng et al., 2023), we calculated the number of MACs associated with these models. As the number of modalities and additional imputation models increases quadratically, so do the associated MACs. This trend underscores the critical need for scalable approaches to support multimodal FL applications efficiently.

**Motivation #1:** Existing FL solutions face inefficiency and scalability challenges as the number of modalities increases.

### A.2 PERFORMANCE DEGRADATION IN EARLY FUSION

In contrast to intermediate fusion and deep imputation, early fusion is far more efficient because it requires training just one model (Snoek et al., 2005). It is particularly advantageous for multimodal time-series healthcare sensing, as it can accurately capture complex inter-modal relationships (Pawłowski et al., 2023). However, it faces challenges when dealing with incomplete modalities in federated learning (FL). Imputing missing modalities using raw data statistics (Zhang, 2016; Van Buuren, 2018) is not feasible in FL environments, as the server lacks access to clients' raw data. As a result, zero-imputation becomes the only viable option (Van Buuren, 2018), leading to performance degradation.

To evaluate the performance with zero-imputation in the absence of modalities, we conducted experiments on two representative mobile sensing datasets, RealWorld (Sztyler & Stuckenschmidt, 2016) and WESAD (Schmidt et al., 2018), each featuring ten modalities, such as accelerometer, gyroscope, temperature, electrocardiogram, electrodermal activity. We simulated conditions where $p\%$ of clients possess incomplete modality data. We allowed $p\%$ of clients (with $40 \le p \le 80$) to randomly omit up to $M-1$ modalities from their training data, where $M$ represents the total number of available modalities.

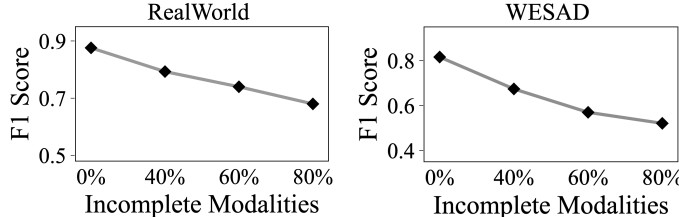

Figure 6: The model performance decreases with more incomplete modalities in RealWorld (left) and WESAD (right) datasets.

Figure 6 shows the average F1 score of a classification task in a respective dataset (activity recognition in RealWorld and stress detection in WESAD) as the number of clients with incomplete modalities increases. The results show a consistent decline in model performance across both datasets, underscoring the negative impact of missing modalities on early fusion. This highlights the need for more carefully designed solutions to handle incomplete modalities in early fusion.

**Motivation #2:** The performance of early fusion with zero imputation gradually worsens as the number of clients with incomplete modalities increases.

## B  IMPACT OF MODALITY-INVARIANT REPRESENTATION LEARNING

To verify the effectiveness of modality-invariant representation learning, we examine the embedding distances and performance (F1 score) of models trained with and without supervised contrastive objective, both of which include cross-entropy loss for classification. The experiment results for PAMAP2 and WESAD datasets are shown in Figure 7. The x-axis represents the number of missing modalities; the left y-axis indicates the distance between complete and incomplete modalities embeddings, and the right y-axis shows the F1 score.

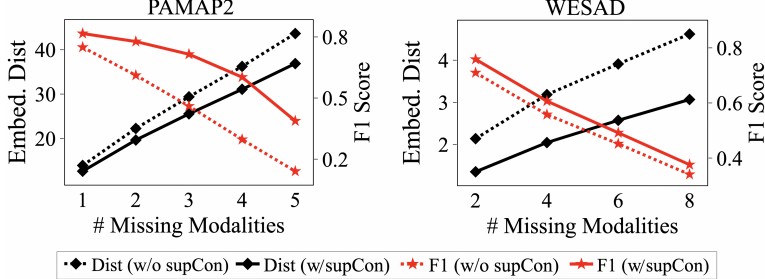

Figure 7: Comparison of models trained with and without supervised contrastive loss for PAMAP2 (left) and WESAD (right) datasets.

We observe that the embedding distance between complete and incomplete modalities consistently rises as the number of missing modalities increases. However, the model trained with supervised contrastive learning can reduce this embedding distance, bringing the embeddings closer to those of the complete data. This suggests that modality-invariant representation learning using a supervised contrastive objective effectively enhances representation learning for incomplete modalities.

## C  ENTROPY TO ASSESS CLIENT MODALITY QUALITY

To evaluate whether entropy can effectively reflect modality information, we conducted experiments logging the model's entropy on training (with incomplete modality) data and the corresponding F1 score on test (with complete modality) data. As we consider all modality combinations per missing modality number, the likelihood of losing more important modalities rises with more missing modalities. This allows us to estimate the impact of the absence of various modality types and numbers. Figure 8 shows the results of experiments with PAMAP2 and WESAD datasets.

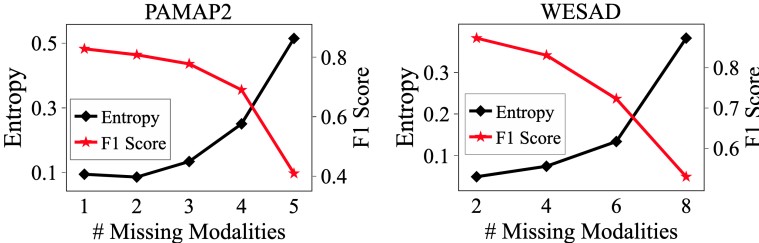

Figure 8: The relationship between entropy and F1 score for PAMAP2 (left) and WESAD (right) datasets.

As the number of missing modalities increases, entropy increases while the F1 score consistently decreases. This indicates that entropy can serve as a proxy to estimate the quality of modalities each client possesses.

## D  EXPERIMENT DETAILS

Below, we describe multimodal time-series healthcare sensing datasets used in our experiments (D.1), baseline methods (D.2), and the implementation details (D.3).

### D.1  DATASETS

Table 4 shows the four real-world datasets used in our experiments: PAMAP2 (Reiss & Stricker, 2012), RealWorld HAR (Sztyler & Stuckenschmidt, 2016), WESAD (Schmidt et al., 2018), and Sleep-EDF (Goldberger et al., 2000; Kemp et al., 2000).

Table 4: Summary of datasets.

| Dataset | #Modalities | Modality Types |
|---------|-------------|----------------|
| PAMAP2 | 6 | Acc, Gyro |
| RealWorld | 10 | Acc, Gyro |
| WESAD | 10 | Acc, BVP, ECG, EDA, EMG, Resp, Temp |
| Sleep-EDF | 5 | EEG Fpz-Cz, EEG Pz-Oz, EOG, EMG, Resp |

**PAMAP2** (Reiss & Stricker, 2012) contains data from nine users performing twelve activities, captured using Inertial Measurement Unit (IMU) sensors. We excluded data from one participant due to the presence of only a single activity data (Jain et al., 2022). The dataset includes readings from accelerometers and gyroscopes modalities, positioned on three different body parts: the wrist, chest, and ankle, resulting in a total of six sensing input modalities.

**RealWorld HAR** (Sztyler & Stuckenschmidt, 2016) (abbreviated as RealWorld), is a human activity recognition (HAR) dataset, collected from fifteen participants performing eight activities. Each participant was equipped with seven IMU devices positioned on seven different body parts, but we omitted two of them due to incomplete activity coverage. Thus, the dataset includes ten modalities from five body locations and two types of IMU sensors.

**WESAD** (Schmidt et al., 2018) is a multi-device multimodal dataset for wearable stress and affect detection. It encompasses data collected from fifteen participants who wore both a chestband and a wristband, capturing physiological sensor data such as Electrocardiogram (ECG), Electrodermal Activity (EDA), Electromyogram (EMG), Blood Volume Pressure (BVP), and Respiration (Resp), Skin Temperature (Temp), in addition to motion data via an Accelerometer (Acc). The objective is to classify the participants' emotional states into three categories: neutral, stress, and amusement. The chestband monitored Acc, ECG, EMG, EDA, Temp, and Resp, whereas the wristband tracked Acc, BVP, EDA, and Temp, collectively resulting in ten distinct modalities.

**Sleep-EDF** (Goldberger et al., 2000; Kemp et al., 2000) comprises sleep recordings from 20 participants. It includes Electroencephalography (EEG), Electrooculography (EOG), chin EMG, Respiration (Resp), and event markers. The labels correspond to five types of sleep patterns (hypnograms).

Similar to previous works (Tsinalis et al., 2016; Phan et al., 2018), we use data from the Sleep Cassette study, which investigated the effects of age on sleep of healthy individuals.

## D.2 BASELINES

**FedAvg** (McMahan et al., 2017) represents the foundational approach to FL, enabling decentralized training without sharing raw data. As a baseline framework, FedAvg is crucial for assessing the lowest achievable accuracy, especially in scenarios lacking specific mechanisms to address missing modalities.

**FedProx** (Li et al., 2020b) was proposed to address system and statistical heterogeneity. It enhances performance by adding a proximal term to the local training loss function to minimize the discrepancy between the global and local models.

**MOON** (Li et al., 2021) targets local data heterogeneity problem. It incorporates contrastive learning into FL to reduce the gap between the global and local model's embeddings while increasing the disparity from the embeddings of the previous local model. MOON has demonstrated superior performance over other FL methods across different image classification tasks, showcasing its effectiveness.

**FedMultiModal** (Feng et al., 2023) (abbreviated as FedMM) is designed to tackle missing modality issues in multimodal FL applications. Initially, FedMM conducts unimodal training for each available modality across all clients. It then merges these unimodal representations through a cross-attention mechanism (Vaswani et al., 2017).

**Harmony** (Ouyang et al., 2023) is proposed to manage incomplete modality data in multimodal FL tasks. It structures the FL training process into two distinct stages: initial modality-wise unimodal training and a second stage dedicated to multimodal fusion. Additionally, Harmony incorporates modality biases in the fusion step to address local data heterogeneity.

**AutoFed** (Zheng et al., 2023) is framework for autonomous driving that addresses heterogeneous clients in FL, including the problem of missing data modalities. AutoFed pre-trains a convolutional autoencoder to impute the absent modality data. The autoencoder is pre-trained using a dataset with complete modality data. However, AutoFed cannot be directly applied to scenarios involving more than two modalities and incurs significant overhead for pre-training with an increasing number of modalities. Therefore, we implemented AutoFed+ to adapt to datasets with more than two modalities (details are provided in Appendix D.4).

## D.3 IMPLEMENTATION DETAILS

We use a 1D convolutional neural network (CNN) as the encoder architecture, following the design for sensing tasks (Haresamudram et al., 2022). For a fair comparison, we standardized the encoder models across all methods. We set random client selection rates from 30% to 50% based on the total dataset users. Standard settings include a learning rate of 0.01, weight decay of 0.001, and a batch size of 32, with SGD as the optimizer. After a grid search to fine-tune the hyperparameters for each baseline, we adjusted the MOON's learning rate to 0.001 and its batch size to 64. We performed all experiments with five seeds and reported the average values.

## D.4 AUTOFED+ IMPLEMENTATION DETAILS

AutoFed+ consists of two phases: (1) pre-training $M(M-1)$ autoencoders and ranking, and (2) the main FL training. Initially, we sample 30% of clients, ensuring they have complete modalities for pre-training purposes. The remaining 70% of the clients participate in the main FL training, with incomplete modality ratios $p$ assigned as in our main evaluation experiments.

During pre-training, the clients' data is further divided into training and validation sets, with the validation set used to rank the imputation models. This ranking is essential because, in the main FL training, if a client $k$ has $m_k$ available modalities and $M - m_k$ missing modalities, then for each missing modality, there are $m_k$ candidate imputation models available to fill the gaps. These models are ranked based on validation loss, computed via distance between original and generated modality data. We perform imputation model pre-training for 50 global rounds with three local epochs and

100 percent client selection rate. Following this, the main FL training is performed for 100 rounds with five local epochs.

For a fair comparison with AutoFed+, as `FLISM` does not include a pre-training phase. Instead, we use the clients that AutoFed+ employed in the pre-training stage to train the FL model for `FLISM` and exclude these clients from the testing phase to maintain consistency with AutoFed+. Similar to AutoFed+, the main FL training is conducted for 100 rounds, with five local epochs per round.

## E ANALYSIS ON COMPLETE MODALITIES

Table 5: Accuracy improvement of `FLISM` over baselines with complete modalities. **EF** denotes Early Fusion, whereas **IF** represents Intermediate Fusion.

| Method | FedAvg (EF) | | FedProx (EF) | | MOON (EF) | | FedMM (IF) | | Harmony (IF) | |
|---|---|---|---|---|---|---|---|---|---|---|
| **Accuracy** | F1 | $\Delta$ F1 | F1 | $\Delta$ F1 | F1 | $\Delta$ F1 | F1 | $\Delta$ F1 | F1 | $\Delta$ F1 |
| | | | | | $p = 0\%$ | | | | | |
| PAMAP2 | .804 | .034 ↑ | .806 | .032 ↑ | .778 | .060 ↑ | .784 | .054 ↑ | .774 | .064 ↑ |
| WESAD | .810 | .016 ↓ | .810 | .016 ↓ | .536 | .258 ↑ | .598 | .196 ↑ | .616 | .178 ↑ |
| RealWorld | .878 | .004 ↑ | .874 | .008 ↑ | .838 | .044 ↑ | .826 | .056 ↑ | .774 | .108 ↑ |
| Sleep-EDF | .706 | .014 ↑ | .704 | .016 ↑ | .714 | .006 ↑ | .638 | .082 ↑ | .586 | .134 ↑ |

Table 5 presents the F1 scores of `FLISM` in comparison to other baselines under complete modality scenario ($p = 0\%$), where all clients have full modalities. The results indicate that `FLISM` outperforms both early and late fusion methods in most cases, demonstrating its effectiveness not only with incomplete modalities but also when all modalities are present.

## F DISCUSSIONS

We outline discussions and promising directions for future research.

**Server Aggregation in Extreme Scenarios.** In §3.2.2, we introduced the Modality Quality-Aware Aggregation (MQAA) to prioritize client updates with high quality modality data. However, in extreme cases where high quality updates are limited or absent, the global model may struggle to generalize, and overemphasis on specific modalities or demographics could lead to unfair outcomes. To address this, MQAA could be extended to include constraints that limit the influence of a single or a group of clients. Additionally, adopting a more refined client selection strategy that ensures diverse client inclusion can prevent overreliance on high-quality clients. Nevertheless, we consider such scenarios unlikely. In our experiments with real-world multimodal healthcare sensing datasets, our method performed effectively, indicating that such extreme cases are rare.

**Extension to High-dimensional Modalities.** Recently, the study of large multimodal models, incorporating modalities such as images and text, has gained significant attention (Radford et al., 2021; Saito et al., 2023). Although `FLISM` performs well in accuracy and system efficiency, it mainly focuses on 1D time-series multimodal sensing applications. This focus stems from observations that early fusion is more efficient than intermediate fusion. We plan to extend `FLISM` to include high-dimensional modalities, such as images and audio. This could involve utilizing small pre-trained models to extract features from these modalities, aligning the extracted features, and proceeding with the `FLISM` training. This approach could broaden the applicability of `FLISM` to a wider range of multimodal integration scenarios, enhancing its versatility and effectiveness.

**Runtime Handling of Incomplete Modalities.** We focused on scenarios involving static modality drops. This approach stems from our observation that dropping the entire modality throughout the FL training causes the highest accuracy degradation. However, we acknowledge that dynamic modality drops, where modalities might become unavailable at various points during application runtime, is also a critical aspect. The Modality-Invariant Representation Learning (§3.2.1), a component of `FLISM`, is designed to accommodate extensions for simulating various dynamic drop scenarios. Further development of the method to specifically cater to dynamic drop scenarios at runtime is an area for future exploration.

**System Heterogeneity-Aware Client Selection.** Although `FLISM` achieves a balance between system efficiency and model accuracy, it overlooks individual user system utilities, such as WiFi connectivity, battery life, and CPU memory. As highlighted in our motivation, the number and type of modalities available for local training vary by user, and the system utilities can change dynamically. Building on the Modality Quality-Aware Aggregation (§3.2.2), we can devise an additional client selection method that accounts for device utility to enhance convergence speed. Future research could explore adapting the method to accommodate system heterogeneity.

