# OpenReview forum: "Federated Learning for Time-Series Healthcare Sensing with Incomplete Modalities"
_ICLR.cc/2025/Conference — ICLR 2025 Conference Withdrawn Submission_

### Official Review · Reviewer_bxfZ · 2024-10-22

**Soundness:** 2
**Presentation:** 3
**Contribution:** 2
**Rating:** 5
**Confidence:** 4

**Summary:**

This paper proposes FLISM to tackle incomplete modalities efficiently. FLISM is an efficient FL training algorithm with incomplete sensing modalities while maintaining high accuracy. In particular, FLISM introduces three techniques to solve the corresponding issues. It utilizes contrastive learning to learn modality-invariant representations, utilizes entropy to weight-prioritize contributions, and utilizes knowledge distillation to reduce local update shifts caused by modality differences.

**Strengths:**

1. The paper is well-organized.
2. The proposed method achieves considerable performance.

**Weaknesses:**

1. The proposed method seems to be general and not particularly related to time-series healthcare sensing. Could you add experimental comparisons with the Harmony method under the settings of the Harmony paper in the corresponding experimental section?
2. The contributions of the proposed method appear insufficient; the combination of missing modalities and the introduced techniques are not particularly strong. For example, entropy may also represent data complexity, so the choice based on entropy might not necessarily be better. Could invariant representation learning based on comparisons lead to decreased discriminative ability, negatively affecting certain modalities, especially the missing ones? Distillation learning could perhaps be interpreted as personalization or something else. The three challenges also lack corresponding experimental evidence and analysis. In summary, the entire method seems to combine existing technologies, resembling a general engineering solution that might still be applicable in other scenarios. It would be best to add some theoretical explanations and connections to incomplete modalities.
3. The compared early fusion methods are relatively old, traditional federated learning approaches unrelated to modality missing. Could you compare some SOTA relevant methods?
4.  When modality data is severely missing, the performance and efficiency seem to be worse than the Harmony method.
5. Is the proposed method effective across different backbones? It would be helpful to supplement results on newer architectures like transformers and Mamba.
6. It would be best to increase analysis to demonstrate the robustness and specificity of the proposed method, such as visual analysis to show that the method genuinely addresses issues caused by modality missing, rather than relying on general tricks, and sensitivity analysis to demonstrate method robustness.

**Questions:**

See above.

---

### Official Review · Reviewer_Gf2U · 2024-10-25

**Soundness:** 2
**Presentation:** 3
**Contribution:** 2
**Rating:** 3
**Confidence:** 5

**Summary:**

This paper aims to address the challenge of missing modalities in multimodal FL frameworks, particularly in cases involving numerous modalities. The paper proposes the FLISM algorithm, which utilizes early fusion in multimodal FL and incorporates three key components. MIRL employs modality dropout to enhance the model’s robustness to missing modalities, MQAA performs weighted aggregation based on client entropy, and GAKD assists local learning through KL divergence. The paper compares FLISM against three SOTA multimodal FL methods and three traditional FL methods in four datasets, considering different modality missing rates, communication, and computation overhead.

**Strengths:**

The paper is well organized, and the proposed methodology is interesting, including modality dropout and entropy in the context of multimodal FL. In addition, the paper conducts extensive comparisons with baselines, and the results show that the proposed method outperforms the baselines.

**Weaknesses:**

- I feel that the level of novelty is somewhat limited. Many elements mentioned in the paper, though applied in a new context of multimodal FL, are existing techniques (e.g., modality dropout). Other aspects, such as early fusion, entropy-based weighted aggregation, and knowledge distillation using KL divergence, are also well established. I can find similar techniques just by randomly searching on Google Scholar.

[1] Improving Unimodal Object Recognition with Multimodal Contrastive Learning

[2] Contrastive Learning based Modality-Invariant Feature Acquisition for Robust Multimodal Emotion Recognition with Missing Modalities

[3] FedGKD: Towards Heterogeneous Federated Learning via Global Knowledge Distillation

[4] FL-FD: Federated learning-based fall detection with multimodal data fusion

- The reduction in the number of parameters, communication, and computation overhead presented in the paper stems from the use of early fusion. This is a contribution of early fusion (and this paper is not the first to apply early fusion in multimodal FL, see [4]) and is not directly related to the methods proposed (i.e., MIRL, MQAA, and GAKD).

- The proposed methods lack generality and are confined to one-dimensional time-series data. The early fusion approach used in the paper requires that the data dimensions of different modalities be consistent. The four datasets selected for evaluation, such as IMU, EMG, and ECG, consist of one-dimensional data vectors. The methods proposed in this paper may not work with data that have varying dimensions (e.g., images), and the paper does not discuss the feasibility or performance of reshaping other types of data.

- The paper does not clarify the roles of entropy in reflecting client modality updates and client modality quality. Is entropy indicating data quality, modality quality, or model update quality? Why not simply calculate the weight based on the number of modalities? Since in Figure 8, they are inversely correlated.

- Typical modality fusion strategies include three approaches, yet the paper neglects late fusion and thus should compare against multimodal FL frameworks that employ late fusion.

**Questions:**

- How are the datasets split for experiments, i.e., do each client correspond to the participants in the datasets? What about the training and test sets? Are there any test users (test clients)?
- There are three definitions of $p$ in the paper. In Eq. (6), it is defined as a probability; in Algorithm 1, it is the fraction of clients; and in the experimental section, it is defined as the incomplete modality ratio.
- Is the incomplete modality of each client dynamically changing during the FL process? For example, could client 1 have modalities A and B in the first communication round, but B and C in the second round?

---

### Official Review · Reviewer_Pwgs · 2024-10-27

**Soundness:** 3
**Presentation:** 2
**Contribution:** 2
**Rating:** 3
**Confidence:** 4

**Summary:**

Briefly summarize the paper and its contributions. This is not the place to critique the paper; the authors should generally agree with a well-written summary.

This paper introduces an efficient federated learning framework specifically designed for federated multimodal sensing applications with incomplete modalities, called FLISM. The authors mainly propose three techniques to address the missing modalities:

1. Modality-invariant representation learning. which uses supervised contrastive learning to extract features robust to missing modalities by augmenting input data through dropping random modalities and adding noises during early fusion.
2. Modality quality aware aggregation that prioritizes high-quality multimodal data from heterogeneous clients.
3. Global aligned knowledge distillation that aligns the local client models with the global server model to reduce biases introduced by limited modalities training on clients.

The authors compare FLISM with different SOTA federated multimodal learning frameworks and demonstrate improvements in F1 scores across different benchmarks. The early fusion also shows greater efficiency compared to the late fusion since only one encoder is involved regardless of the number of modalities. The authors also evaluate FLISM with an imputation method (AutoFed) to handle incomplete data and demonstrate significant improvements in performance and computational costs.  Ablation studies are also conducted to understand the performance of FLISM.

**Strengths:**

- The challenge described is clear and meaningful for multimodal learning.
- The design is well supported by the motivation and address the concern raised in the challenge.
- The authors provide detailed illustrations and easy-to-follow algorithm to clarify their technical methods.
- The method demonstrates both superior performance and significant efficiency gains compared to SOTA frameworks.

**Weaknesses:**

**Major**

- I am concerned with the novelty of the work. There have been many works on modality dropout to improve multimodal robustness. Although some of these works may be conducted in the representation space, they share the same concept of randomly dropping a subset of the modalities to improve robustness. Furthermore, some previous works have also performed knowledge distillation [1] to regularize the discrepancy between the clients. The authors should expand the related works, discuss how the proposed method is different from existing work, and emphasize the key novelties.
- The authors should improve the evaluation presentation to reduce confusion. For example, the term $p$ is used in multiple ways where p means incidence of clients with incomplete modalities for 4.2 but in later sections it represents the ratio of modalities that is missing.
- The authors should also elaborate on the experimental setup. Are the incomplete modality ratios for the training data (from the appendix)? Then, what is the incomplete modality ratio for the testing data? If the testing data is complete, what is the performance like if it contains incomplete multimodal data, which is common under the sensor deployment setting?
- The number of nodes involved for each dataset should also be specified.
- The efficiency in communication and computation cost reflects the advantage of FLISM over methods that leverage intermediate fusion. It does not show a comparison with any early fusion frameworks. Early fusion frameworks naturally have lower costs and a lower number of parameters as the number of modalities scales, so it may not be a fair evaluation if Figure 2 excludes those early fusion frameworks.
- The authors could clarify how they perform the fusion. It is unclear how the early fusion is performed. How does this generalize to the case when the time-series input modalities have different sampling rates, which could lead to different information densities? The performance of additional datasets with more diverse modalities could improve the significance of the work.

**Minor**

- Some parts of Table 1 are a bit confusing. The caption and the left column say accuracy, but the indicators in each column say F1 and delta F1.
- Appendix 5. E: Late fusion → Intermediate fusion.
- Since the setup is unclear and the code has not been released, the reproducibility of this work is also unclear.

[1] FedGKD: Toward Heterogeneous Federated Learning via Global Knowledge Distillation

**Questions:**

See weakness for more details.

- How is the fusion performed? How is the heterogeneity in data structured and sampling rate handled? Please elaborate and see weakness for more details.
- What are the major novelties compared to previous modality dropout and global-local knowledge distillation methods?
- Why is the same modality from different locations considered different? (e.g. three same modalities from 2 different locations == 10 modalities) Please compare how this is treated with previous works (e.g. Harmony).

---

### Official Review · Reviewer_41iq · 2024-11-02

**Soundness:** 3
**Presentation:** 4
**Contribution:** 4
**Rating:** 6
**Confidence:** 3

**Summary:**

This paper tries to develop an efficient and scalable federated learning algorithms that handles incomplete modalities while maintaining high accuracy. Healthcare applications increasingly reply on multimodal data and there is need to utilize such data in a privacy preserving manner (.e.g, FL). But incomplete modalities brings challenges to FL. Deep imputation methods are unrealistic in FL setting and not scalable as the number of modalities increases. Intermediate fusion methods scales poorly with number of modalities as well, also incurs high communication and computation costs. Early fusion is more efficient but may lead to performance degradation due to drift etc.

The authors proposed FLISM, an early fusion based algorithm with 3 key components to address the challenges above. Modality-Invariant Representation Learning use supervised contrastive learning to learn a representation that's consistent despite varying input modalities. This addresses early fusion with zero imputation may lead to biased modality relationships. Modality Quality-Aware Aggregation uses an entropy-based metric to evaluate and weights updates from clients. Global-Aligned Knowledge Distillation use KD to align local model with global model to enhance model's ability to generalize over different modality sets.

The proposed algorithm is evaluated on 4 datasets and compared against state of the art baselines. The proposed algorithm shows good performance and system efficiency.

**Strengths:**

- The paper is well motivated and address an important real-world problem in FL (modality variance).
- The early fusion with zero imputation fits well in the FL paradigm. The proposed algorithm provides mechanism at both client and server end to consider the modality variance problem.
- The proposed solution is evaluated on 4 different datasets and on both model and system performance metrics.
- The paper is well organized and written. The presentation is clear and easy to follow.

**Weaknesses:**

- Although assumed a multi-modal setting, the solution is only evaluated on time series data.
- The proposed solution is only evaluated under static modality drop. Whereas in reality, the modality variance is more dynamic.
- Although the authors provided empirical evidence to support entropy-based metric for client selection, they only seem to consider model performance and did not provide much discussion on fairness.

**Questions:**

- How generalizable the proposed solution is to other modalities such as image and audio?
- Can you describe how the modality variance could potentially be handled?
- Will the Modality Quality-Aware Aggregation method cause biases towards users/device with more modality and hence cause performance drop for other users?

---

### Note · Authors · 2024-11-14

I have read and agree with the venue's withdrawal policy on behalf of myself and my co-authors.